# LEARNING MUSIC STYLE FOR PIANO ARRANGEMENT THROUGH CROSS-MODAL BOOTSTRAPPING

## ABSTRACT

What is music style? Though often described using text labels such as "swing," "classical," or "emotional," the real style remains implicit and hidden in concrete music examples. In this paper, we introduce a cross-modal framework that learns implicit music styles from raw audio and applies the styles to symbolic music generation. Inspired by BLIP-2, our model leverages a Querying Transformer (Q-Former) to extract style representations from a large, pre-trained audio language model (LM), and further applies them to condition a symbolic LM for generating piano arrangements. We adopt a two-stage training strategy: contrastive learning to align auditory style with symbolic expression, followed by generative modelling to perform music arrangement. Our model generates piano performances jointly conditioned on a lead sheet (content) and a reference audio example (style), enabling controllable and stylistically faithful arrangement. Experiments demonstrate the effectiveness of our approach in piano cover generation, style transfer, and audio-to-MIDI retrieval, achieving substantial improvements in style-aware alignment and music quality.[1]

## 1    INTRODUCTION

Automatic music generation is often controlled by *explicit* content such as melody, chords, and text labels (Yang et al., 2019; Wang et al., 2020b; Lu et al., 2023; Bhandari et al., 2025), but music concepts can be more nuanced than we often realize. When musicians learn a style, instead of relying on abstract descriptors like "romantic" or "jazz" alone, they absorb patterns from music examples that share common stylistic traits. The commonality across these examples forms a style, an *implicit* one that cannot be fully described with words or labels but only understood through the music itself. This paper studies such implicit style qualities, particularly regarding *grooving patterns and dynamics* in piano performances, where abstract descriptors often oversimplify their richness, even though they are immediately perceivable in audio. We explore how such implicit style can be internalized from given audio examples and control music generation in a deep learning framework.

Large-scale music language models (music LMs) have shown strong capabilities in learning explicit music content, as demonstrated by probing studies (Wei et al., 2024; Ma & Xia, 2024; Ma et al., 2024; Vásquez et al., 2024; Castellon et al., 2021) and adapter-based designs (Lin et al., 2024a; Zhang et al., 2024; Lin et al., 2024b; Wu et al., 2024). Yet, control over implicit style remains limited. For example, when using audio to guide symbolic music generation, existing models can extract melody and chords (Donahue et al., 2022; Wang et al., 2022), but capturing stylistic traits like the rhythmic feel and more expressive nuances remains a greater challenge. This requires disentangling style from music content, which current music LM-based studies have yet to explore.

In this paper, we explore learning implicit music style in a cross-modal setting for symbolic piano arrangement. Our goal is to generate an expressive piano MIDI performance conditioned on two inputs: an audio example (providing style) and a lead sheet MIDI score (melody and chords as content). To achieve this, we connect pre-trained music LMs in the audio and symbolic domains using a Querying Transformer (Q-Former), a lightweight Transformer originally designed for vision-language alignment (Li et al., 2023). As shown in Figure 1, we extend its role to capture implicit music style, extracting a *style representation* from the hidden states of an audio LM. Then, a

---

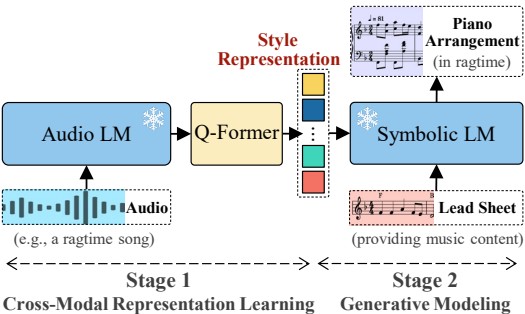

Figure 1: A Q-Former module bridges the modality gap between a frozen audio LM and a symbolic music LM. It extracts cross-modal music *style* from the hidden representations of the audio LM and, together with a lead sheet providing music *content*, conditions the symbolic LM for piano arrangement. The Q-Former is trained in a two-stage process, effectively bootstrapping audio-to-symbolic arrangement without re-training either LM backbone.

symbolic LM conditions on this representation, along with the *content* of the lead sheet, to generate a piano arrangement. The Q-Former enables cross-modal style transfer between two large unimodal LMs without re-training them—a process we refer to as *bootstrapping*.

In our design, we treat the Q-Former as a bottleneck to transfer only style-related information and adopt a two-stage training strategy. The first stage employs contrastive learning, training the Q-Former to extract auditory representations that are musically relevant, expressible in symbolic piano arrangements, and independent of explicit music content. These include accompaniment textures, grooving patterns, and performance dynamics (MIDI velocity contour and tempo). The second stage focuses on generative modeling, where the Q-Former's output conditions the symbolic LM to generate a piano arrangement with the desired expression. We show that the complete system generates more stylistically accurate cover songs compared to existing audio-to-symbolic arrangement methods, while also enabling piano style transfer by conditioning on alternative audio examples. In addition, we demonstrate that the Q-Former can also be applied to audio-to-symbolic retrieval, further highlighting its strength as a general-purpose cross-modal representation learner.

In summary, the contributions of this paper are threefold:

1. We use the Q-Former to **align audio and symbolic modalities through implicit music style**, extending its role beyond content alignment in vision-language tasks.

2. We present **a new methodology to disentangle music style from large, pre-trained LMs**, offering a more scalable alternative to traditional latent-variable disentanglement methods.

3. Our model achieves **style-aware audio-to-symbolic piano cover arrangement**. Experiments demonstrate that it outperforms existing audio-to-symbolic models, including both disentanglement-based methods and standard LM approaches.

## 2 RELATED WORKS

We review two areas of key relevance. Section 2.1 discusses recent advances in music LMs. Section 2.2 further introduces piano cover generation, the primary task addressed in this paper.

### 2.1 MUSIC LANGUAGE MODELS

Rapid progress in large-scale language models has transformed how we interact with various forms of media, including text, image, and music (Touvron et al., 2023; Alayrac et al., 2022; Li et al., 2023; Kong et al., 2024; Yuan et al., 2025). In particular, large music LMs (Agostinelli et al., 2023; Melechovský et al., 2024; Thickstun et al., 2024; Bhandari et al., 2025) have notably influenced creative practices and user experiences. Models like MusicGen (Copet et al., 2023) can generate music audio with rich timbres directly from text, while MuseCoco (Lu et al., 2023) produces symbolic compositions with well-structured textures in varied genres. These advancements are driven

by training large-scale neural networks on extensive data, scaling up to billions of model parameters to enhance controllability and musicality.

Despite these successes, most existing music LMs operate in an unimodal setting, focusing solely on either audio or symbolic representations. Although text-to-music generation has been increasingly effective (Agostinelli et al., 2023; Melechovský et al., 2024; Bhandari et al., 2025), text descriptions may fall short in expressing nuanced style or performance subtlety. In contrast, our work explores a cross-modal framework that bridges audio and symbolic modalities. By leveraging the strong perceptual understanding of audio LMs and the expressive composition capabilities of symbolic LMs, we bootstrap a system for audio-to-symbolic arrangement. This approach enables more intuitive and fine-grained control over music style beyond what can be conveyed through text alone.

### 2.2 PIANO COVER GENERATION

Piano cover generation aims to reinterpret an audio recording as a symbolic piano performance. Unlike traditional music transcription, which primarily analyses note-level content such as pitch and timing (Kong et al., 2021; Ou et al., 2022; Gardner et al., 2022; Gu et al., 2024; Zeng et al., 2024), a piano cover often targets higher-level, more structured music elements that shape the *feel* of a performance. The goal is to generate symbolic arrangements that not only sound correct but also feel musically aligned with the original audio.

Existing approaches to piano cover generation often leverage pre-trained transcription models, which primarily extract melodic and harmonic content from the audio (Nakamura & Yoshii, 2018; Tan et al., 2024b;a; Wang et al., 2022; Choi & Lee, 2023). However, such models tend to overlook stylistic nuances, resulting in outputs accurate in harmony but lacking the expressive character of the source performance. In this paper, we re-frame piano cover generation through the lens of content-style disentanglement, acquiring *content* in the symbolic form (i.e., melody and chord progression) while learning *style* from the audio. This approach bridges the audio-symbolic gap more effectively, capturing not just *what* is played, but *how* it is played.

## 3 METHOD

Our goal is to learn music style representations from the audio and leverage these representations to arrange symbolic piano performances. To bridge the modality gap from audio to symbolic music, we adopt the Q-Former (Li et al., 2023) under a two-stage training strategy. As shown in Figure 2, Stage-I focuses on audio-symbolic representation learning with a frozen audio LM, while Stage-II addresses audio-to-symbolic arrangement with a symbolic LM. In Section 3.1, we first introduce our audio-symbolic data pairing method that facilitates style learning. We illustrate the Q-Former architecture in Section 3.2, followed by the two-stage training procedure in Sections 3.3 and 3.4.

### 3.1 DATA PAIRING FOR STYLE LEARNING

Training an audio-to-symbolic alignment model requires paired audio–MIDI data. In this work, we use 10-second audio clips paired with 4-bar MIDI segments. Since our goal is to capture style rather than low-level note transcription, we construct the pairs to be *loosely aligned*. Specifically, for each audio clip, we select a MIDI segment near its center with a random temporal shift of up to $\pm 1$ second, and we randomly transpose the MIDI into all 12 keys. This design assumes that music style is locally consistent, while discouraging the model from memorizing exact note-to-note correspondences. In the following sections, we show that the two-step training method enables the model to abstract style features that are shared between the modalities.

We represent music audio as raw waveforms sampled at 32kHz. MIDI is tokenized into note event sequences quantized at 1/12-beat resolution. We include various symbolic features including time signature (quadruple and triple meters), tempo curve, note pitch, duration, and velocity.

### 3.2 Q-FORMER ARCHITECTURE

The Q-Former is a Transformer encoder with two parallel, modality-specific streams that share the self-attention layers. As shown in Figure 2a, it accepts audio and symbolic inputs and learns a cross-

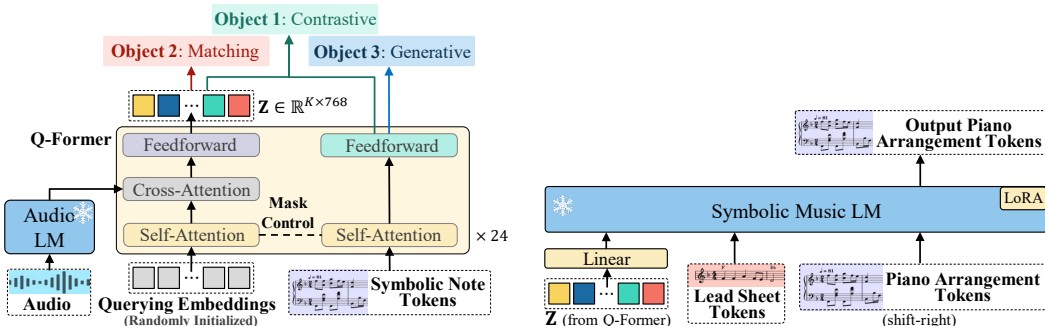

(a) Stage-I: Cross-modal learning with Q-Former.    (b) Stage-II: Audio-to-symbolic arrangement.

Figure 2: Overview of the two-stage framework. (Left) Stage-I: The Q-Former is a Transformer encoder with two parallel, modality-specific streams (audio and symbolic). Both streams are used during training for cross-modal alignment, while only the audio stream is retained at test time. It extracts a cross-modal music style representation $\mathbf{Z}$ via cross-attention to an audio LM. (Right) Stage-II: A symbolic music LM further generates piano arrangements conditioned on (i) the style embedding $\mathbf{Z}$ obtained from the Q-Former and (ii) a lead sheet providing music content.

modal music style representation $\mathbf{Z}$. The left stream is connected to the audio LM via cross-attention, while the right stream encodes symbolic piano arrangement tokens via the shared self-attention. A set of query embeddings (*queries*), initialized randomly, is fed into the left stream and trained to capture style cues from the audio modality while being aligned with the symbolic modality. At test time, only the left stream is retained to uncover cross-modal style directly from audio.

We initialise the Q-Former weights using the pre-trained MusicBERT-Base model (Zeng et al., 2021). The added cross-attention layers are randomly initialized. Following Blip-2 (Li et al., 2023), we use 32 learnable queries, each with dimension 768. Symbolic piano arrangements are tokenized in the OctMIDI format (Zeng et al., 2021), which produces note-wise joint embeddings. Overall, the Q-Former comprises 186M parameters, including the learnable queries and note embedding layers.

### 3.3 STAGE-I: AUDIO-SYMBOLIC REPRESENTATION LEARNING

We integrate the Q-Former into MusicGen (Copet et al., 2023), one of the leading audio LMs available today. The queries interact with audio hidden states through cross-attention while remaining connected to the symbolic stream via shared self-attention layers. However, relying solely on loosely aligned data is insufficient for learning robust style representations, since no direct *content*-level correspondence exists between the audio and symbolic streams (see Section 3.1). To prevent representation collapse and encourage meaningful style abstraction, we introduce three complementary training losses, each paired with a tailored self-attention mask that regulates cross-modal interactions. These objectives are illustrated in Figure 2a and detailed below.

The primary objective is **Audio-Symbolic Contrastive Learning**, which enforces a higher audio-symbolic similarity for positive (original) pairs compared to negative ones (i.e., randomly paired audio and MIDI clips). Let $\mathbf{Z} \in \mathbb{R}^{32 \times 768}$ be the query outputs from the audio stream of Q-Former, and $t \in \mathbb{R}^{1 \times 768}$ be the output embedding of the start token () from the symbolic stream. We define the audio-symbolic similarity as $\max_k(\cos(\mathbf{Z}_k, t))$ for $k = 1, 2, \cdots, 32$, where $\cos(\cdot, \cdot)$ denotes the cosine similarity. The contrastive loss pulls closer aligned audio and symbolic clips in the representation space, while pushing apart unrelated pairs. To prevent information leakage, we employ a *unimodal self-attention mask*, ensuring queries and symbolic tokens do not attend to each other. For detailed mask design, we refer readers to BLIP-2 (Li et al., 2023).

The second objective is **Audio-Symbolic Matching**. It is formulated as a binary classification task, where the model predicts whether a given audio-symbolic pair corresponds to each other. On top of contrastive loss, the matching loss aims to capture a finer cross-modal correspondence. In this case, we apply *no masking*, allowing the queries to attend across modalities. Each query output $\mathbf{Z}_k$ is fed into a binary linear classifier to produce a logit, and the logits from all queries are averaged to

compute the final matching score. To create informative negative pairs, we employ the hard negative mining strategy in (Li et al., 2021).

The final objective, **Audio-Grounded Symbolic Generation**, enforces the Q-Former's right stream to auto-regressively reconstruct the input piano arrangement. We implement a *cross-model causal self-attention mask*, allowing the symbolic tokens to attend to the queries but not vice versa. This objective ensures that the style cues extracted from the audio are sufficiently informative to support symbolic realization. To signal a decoding task, we replace the starting  token with a special <DEC> token. Additionally, we prepend a sequence of lead sheet tokens before the <DEC> token so that the queries are encouraged to extract *style*, rather than transcribing *content* from the audio.

### 3.4 STAGE-II: AUDIO-TO-SYMBOLIC GENERATIVE MODELING

In the generative modelling stage, we take advantage of the generative capability of MuseCoco (Lu et al., 2023), a large-scale symbolic music LM. As illustrated in Figure 2b, MuseCoco is used to reconstruct a piano arrangement based on two concatenated conditional inputs: 1) the query output embeddings $\mathbf{Z}$ from the Q-Former, and 2) a lead sheet. The Q-Former is pre-trained at Stage-I to extract cross-modal music style from the audio, thus providing style guidance. The lead sheet defines the theme melody and chord progression as the content.

To enable compatibility with MuseCoco, we project $\mathbf{Z}$ into the same embedding dimension as MuseCoco's token embeddings via a linear layer. Symbolic note tokens are converted into the REMI format (Huang & Yang, 2020). Despite the slightly different tokenizations used across stages, we find that the latent representations learned in Stage-I remain compatible with Stage-II. Since MuseCoco does not natively support lead sheet conditioning, and the inclusion of the lead sheet tokens alters its input format, we insert a LoRA adapter (Hu et al., 2022) of rank 16 into each self-attention layer. This enables the model to reweight attention and incorporate the new conditioning inputs, while keeping MuseCoco's original parameters frozen.

## 4 ARRANGEMENT DEMONSTRATION

In this section, before presenting experiments, we first demonstrate the performance of our audio-to-symbolic arrangement model under *freely manipulated* audio style references. Figure 3a shows an 8-bar lead sheet excerpt from the musical *The Sound of Music*. The selected passage features harmonically rich chords, including diminished and seventh chord qualities, which present suitable complexity for arrangement experiments. Figures 3b to 3d showcase the arrangement results conditioned on varied audio references. The 8-bar arrangement is generated using windowed sampling, wherein a 4-bar context window progresses forward every 2 bars and continues sampling conditioned upon the preceding 2 bars.

Figure 3b shows the piano cover from the original *The Sound of Music* soundtrack,[2] which features lush orchestration dominated by string ensembles. Our arrangement captures this orchestral essence through dense, block-chord voicing that emulates the sonority of string sections. Additionally, ornaments such as arpeggios and trills are found to complement the sweeping harmonic textures, which contributes to the music's free-flowing character.

Figure 3c shows an arrangement conditioned on the ragtime classic *The Entertainer*.[3] Following the audio recording, the arrangement's tempo is "not fast," and the piano texture distinctly adopts a ragtime rhythm, with steady bass notes on downbeats and syncopated chordal accents on upbeats.

Figure 3d shows an arrangement conditioned on the bossa nova piece *The Girl from Ipanema*.[4] In this interpretation, the arrangement is characterized by a moderate tempo and distinctive left-hand syncopated patterns characteristic of the bossa nova genre.

Across all three piano arrangements, while distinct music styles are effectively captured from the audio references, the theme melody and harmonic structures remain faithfully preserved. In Figure 3, we highlight melody notes preserved from the lead sheet using blue note heads.

---

[2]Original audio: `https://youtu.be/6f0T6UV-HiI&t=57`

[3]Ragtime audio: `https://youtu.be/jKlfNfRZL9I&t=11`

[4]Bossa nova audio: `https://youtu.be/DvA_wDOVD10&t=12`

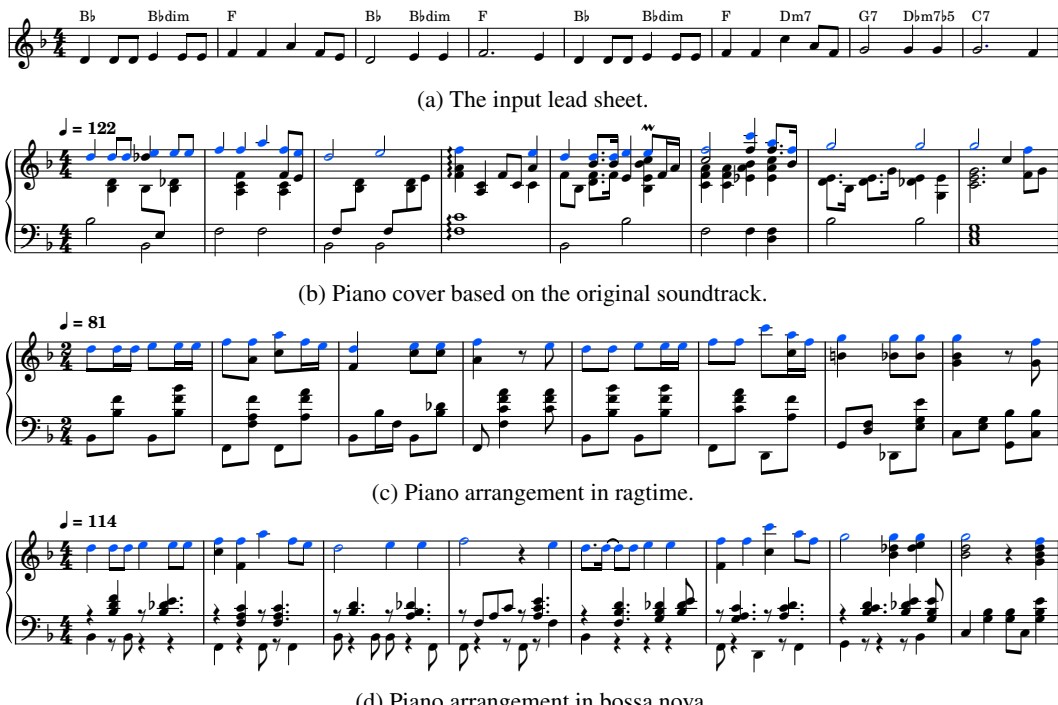

(a) The input lead sheet.

(b) Piano cover based on the original soundtrack.

(c) Piano arrangement in ragtime.

(d) Piano arrangement in bossa nova.

Figure 3: Audio-to-symbolic arrangement for an 8-bar excerpt from *The Sound of Music*. Score is manually engraved from MIDI for visualisation purpose. Figures 3b to 3d are arranged based on the lead sheet in 3a and an audio reference from the original soundtrack, a ragtime piece, and a bossa nova piece, respectively. Preserved music contents are highlighted in blue note heads. Synthesized audio is provided on the demo page: `https://anonymous55aht.github.io/`.

Additional examples are available on our demo page, including arrangements in a wider range of classical, jazz, and pop styles applied to well-known lead sheets, illustrating the flexibility of both style and lead sheet control.

## 5 EXPERIMENTS

This section evaluates our audio-to-symbolic arrangement model. Our model generates piano performances jointly conditioned on a lead sheet and an audio reference. When the two inputs are aligned with each other, the task corresponds to *piano cover generation*; when they are unpaired, the task becomes *cross-modal style transfer*. In this section, we primarily frame our evaluation as *audio-conditioned piano cover generation*, focusing on whether it can transfer audio-derived style while preserving thematic content, which allows direct comparison with prior baselines. For the style transfer setting, we additionally compare with ablation variants of our model in Appendix C.

In Section 5.1, we introduce the datasets used in the experiments. In Section 5.2, we describe the baseline models used for comparison. Our evaluation is divided into two parts: objective evaluation in Section 5.3, and subjective evaluation in Section 5.4. Finally, Section 5.5 extends the evaluation to audio-to-symbolic retrieval, highlighting the Q-Former's broader capability in cross-modal alignment. More details for model configuration and training are covered in the Appendix A.

### 5.1 DATASETS

Our model is trained on two dual-modal datasets: POP909 (Wang et al., 2020a) and PIAST (Bang et al., 2024). POP909 contains 1K piano cover arrangements created by professional musicians. The music genre is primarily Mandarin pop, while the accompanying audio features diverse band instrumentation, which can help the model learn generalizable audio representations of pop music.

PIAST, on the other hand, contains 8K piano recordings along with symbolic transcriptions across a variety of genres, including pop, jazz, and classical. Despite the lack of band instrumentation in the piano recording, the genre diversity of PIAST encourages the model to produce more expressive and stylistically varied performances. We split both datasets at song level into training (90%), validation (5%), and test (5%) sets. Each symbolic MIDI file is clipped into 4-bar segments with a 2-bar hop size, transposed to all 12 keys, and center-aligned with the corresponding 10s audio clip.

We also test on two out-of-domain datasets: Ballroom (Gouyon et al., 2006; Krebs et al., 2013) and GTZAN (Tzanetakis & Cook, 2002; Marchand & Peeters, 2015), both featuring diverse band and orchestral instrumentation, as well as fine-grained music genres such as jive and bossa nova. Since they lack paired symbolic annotations, we use them for testing only. This allows us to assess the model's generalization ability and its capacity to accommodate styles beyond pop music.

## 5.2 BASELINE MODELS

We compare our model against two representative piano cover generation models: *PiCoGen2* (Tan et al., 2024a) and *Audio-to-MIDI* (Wang et al., 2022), as well as one ablation variant of our method.

**PiCoGen2** (PCG2) is a Transformer-based language model that builds on the hidden states of Sheetsage (Donahue et al., 2022; Donahue & Liang, 2021), which itself is derived from Jukebox (Dhariwal et al., 2020), a pre-trained, large-scale music language model. Leveraging Jukebox's internalized understanding of music *content*, PiCoGen2 generates symbolic piano covers directly from audio.

**Audio2MIDI** (A2M) is an auto-encoder-based disentanglement framework, using separate modules to extract beat and tempo (Böck & Davies, 2020), chord (Jiang et al., 2019), and piano texture from audio. The texture extractor is initialized from a piano transcription model (Hawthorne et al., 2018). The extracted components are then merged to form a symbolic piano arrangement.

**Ours w/o Pre-Training** (w/o PT) is an ablation variant of our model in which the Q-Former is trained directly in Stage-II, without undergoing the representation learning phase in Stage-I. This setup tests the validity of the two-stage training strategy we applied in this work.

To ensure a fair comparison with baseline models, we use Sheetsage (Donahue et al., 2022; Donahue & Liang, 2021) to transcribe lead sheets from the audio, making audio the sole input for all methods.

## 5.3 OBJECTIVE EVALUATION

Piano cover generation transforms an audio input into a symbolic piano performance, which should ideally capture not only *what* is played, but also *how* it is played. In Section 5.3.1, we evaluate *content preservation* (*what* is played) and *style coherence* (*how* it is played) using statistical metrics. We hypothesize that stronger style capture can improve overall arrangement quality. Hence, in Section 5.3.2 we further assess *broader audio-to-symbolic coherence* using latent embedding measures.

### 5.3.1 CONTENT PRESERVATION AND STYLE COHERENCE

We first evaluate how well the generated piano cover inherits audio-derived style while preserving thematic content. We introduce two categories of metrics: 1) *content preservation*, and 2) *style coherence*. *Content preservation* assesses whether the melody and harmony are well-maintained. We use *Melody Chroma Accuracy (MCA)* (Choi & Lee, 2023; Tan et al., 2024a) and *Chord Accuracy (CA)* (Ren et al., 2020; Zhao et al., 2024) to measure these aspects. *Style coherence* evaluates whether the accompaniment grooves and performance dynamics are well captured and manifested in the symbolic arrangement. We introduce three metrics: *Grooving Pattern Coherence (GPC)* (Wu & Yang, 2020), *Velocity Contour Coherence (VCC)*, and *Tempo Accuracy (TA)*. Among them, *GPC* and *VCC* compare the generated covers to human arrangements (available for POP909). *TA* compares the generated tempo to the ground-truth (estimatable from audio using (Böck & Davies, 2020) when not annotated). Taken together, these metrics indicate how well the generated cover captures the stylistic "feel" of the reference audio. Across the content and style metrics, a higher value indicates better coherence. We provide more detailed definitions for each metric in Appendix B.

We consider two evaluation settings: 1) *in-distribution* evaluation on the POP909 test set, and 2) *out-of-distribution* evaluation on 100 tracks randomly drawn from the Ballroom and GTZAN datasets.

Table 1: Objective evaluation on *content preservation* and *style coherence*. Ballroom/GTZAN are out-of-distribution sets to assess generalization to unseen genres and instrumentation. Results are reported in the form of mean ± sem$^s$ (in percentage), where sem is the standard error of mean. Different superscript letters $s$ within a column indicate significant differences (p < 0.05/6) based on Wilcoxon signed rank test with Bonferroni correction.

| | POP909 (In-Distribution) Test Set | | | | | Ballroom/GTZAN | | |
| | MCA ↑ | CA ↑ | GPC ↑ | VCC ↑ | TA ↑ | MCA ↑ | CA ↑ | TA ↑ |
|---|---|---|---|---|---|---|---|---|
| **Ours** | $32.0\pm0.5^b$ | $33.3\pm1.1^b$ | $\mathbf{79.2}\pm0.5^a$ | $\mathbf{76.6}\pm0.5^a$ | $\mathbf{83.6}\pm1.5^a$ | $\mathbf{17.8}\pm0.3^a$ | $\mathbf{16.3}\pm0.4^a$ | $\mathbf{79.4}\pm1.1^a$ |
| **PCG2** | $\mathbf{39.3}\pm0.6^a$ | $\mathbf{40.0}\pm0.9^a$ | $78.4\pm0.6^a$ | $73.2\pm0.5^b$ | $74.4\pm2.0^c$ | $17.2\pm0.3^a$ | $15.5\pm0.5^b$ | $57.6\pm1.5^c$ |
| **A2M** | $15.0\pm0.3^c$ | $22.4\pm0.9^d$ | $69.0\pm0.7^b$ | $68.6\pm0.8^c$ | $77.8\pm1.4^b$ | $10.9\pm0.2^b$ | $14.8\pm0.5^b$ | - |
| **w/o PT** | $30.0\pm0.6^b$ | $28.8\pm1.0^c$ | $78.6\pm0.5^a$ | $76.3\pm0.5^a$ | $68.6\pm1.9^d$ | $17.2\pm0.3^a$ | $15.0\pm0.4^b$ | $74.5\pm1.3^b$ |

Table 2: Objective evaluation on *broader audio-to-symbolic coherence* using CLaMP3.

| | POP909 | | Ballroom/GTZAN | |
| | Acc@5 (%) ↑ | Rank ↓ | Acc@5 (%) ↑ | Rank ↓ |
|---|---|---|---|---|
| **Ours** | $\mathbf{27.1}\pm1.2$ | $\mathbf{16.3}\pm0.3$ | $21.4\pm0.6$ | $\mathbf{30.2}\pm0.3$ |
| **PCG2** | $23.6\pm1.3$ | $16.7\pm0.3$ | $\mathbf{21.7}\pm0.9$ | $31.3\pm0.6$ |
| **A2M** | $18.4\pm1.5$ | $18.2\pm0.3$ | $19.6\pm0.8$ | $30.6\pm0.4$ |
| **w/o PT** | $24.2\pm1.1$ | $16.9\pm0.3$ | $18.1\pm0.1$ | $33.1\pm0.5$ |

We run our method and baseline models at each test piece in 10 independent rounds, deriving 450 sets of piano cover samples for POP909, and 1000 sets for Ballroom/GTZAN. We report the mean and standard error. As shown in Table 1, while our method leads in most metrics, we find that on POP909, it achieves a lower *content preservation* score (*MCA* and *CA*) than the *PCG2* baseline. This is probably because our arrangement process relies on Sheetsage-extracted lead sheets, where wrongly estimated chords or melody notes at this stage can propagate to our final results. However, on the out-of-distribution Ballroom/GTZAN dataset, our model surpasses *PCG2* in *MCA* and *CA*, suggesting that *PCG2* is more tailored to pop music (where chords and melodies are relatively simple), whereas our approach generalizes better across diverse music genres and instrumentations. In terms of *style coherence*, our model outperforms all baselines in *GC* and *VC* on POP909, and in *TA* on both test sets, indicating that it can better capture the stylistic characteristics from the audio.

### 5.3.2 BROADER AUDIO-TO-SYMBOLIC COHERENCE

We further evaluate *broader audio-to-symbolic coherence* using CLaMP3 (Wu et al., 2025), which offers a shared latent space bridging the two modalities. We use CLaMP3 as a cross-modal retriever: for each test audio, it computes the cosine similarity between the audio's embedding and all symbolic covers' embeddings. If the most similar symbolic candidate matches the one generated from that audio input, we count it as a correct match, indicating coherent audio–symbolic alignment. Based on this setup, we report two metrics: *Top-5 Retrieval Accuracy (Acc@5*, higher is better), and *Mean Rank*: the average rank position of the correct audio–symbolic pair in all candidates (lower is better).

We evaluate on POP909 and Ballroom/GTZAN, respectively. POP909 provides 45 test pop tracks, for which we use randomly cropped 16-bar excerpts. Ballroom/GTZAN offers 100 tracks spanning more diverse genres and instrumentation, and we evaluate using the full 30s clips. In each run, every model generates one cover per test piece. We repeat the evaluation over 10 independent runs and report the mean and standard error. As shown in Table 2, our model consistently outperforms all baselines, including *PCG2*, on POP909 by a clear margin. This suggests that stronger style capture can lead to higher overall arrangement quality. On Ballroom/GTZAN datasets, our model also attains a substantially lower *Mean Rank*, indicating strong generalization across styles and genres. In contrast, the *w/o PT* variant outperforms baselines on POP909 but fails on Ballroom/GTZAN, confirming that the Stage-I pre-training is crucial for effective cross-modal alignment.

### 5.4 SUBJECTIVE EVALUATION

We further conduct a double-blind online listening survey to evaluate the music quality. The survey comprises 6 test pieces of varied genres drawn from the Ballroom and the GTZAN datasets. Each

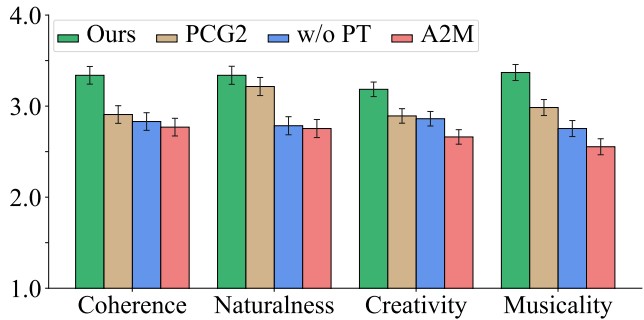

Figure 4: Subjective evaluation on music quality

test piece is accompanied by 4 piano covers interpreted by our model and each baseline model. For each model, we select the best result from 3 generated samples. The purpose is to prevent occasional low-probability failures (e.g., incomplete or degenerate generations) from disproportionately influencing the subjective assessment. All samples are 16 bars long and rendered to audio using the Cakewalk TTS-1 soundfont, resulting in approximately 40s of audio per sample. Both the order of the test pieces and the order of samples are randomized. Participants are asked to complete 3 test pieces by rating each piano cover on a 5-point Likert scale across 4 criteria: 1) *Audio-to-Symbolic Coherence*, 2) *Naturalness*, 3) *Creativity*, and 4) *Overall Musicality*. See more detail in Appendix D.

A total of 21 participants with diverse musical backgrounds completed our survey. The average completion time is 12 minutes. Figure 4 shows the mean ratings and standard errors analyzed using within-subject ANOVA (Scheffe, 1999). The results reveal significant main effects ($p < 0.05$) across all evaluation criteria. While our model performs comparably to the state-of-the-art *PCG2* in *Naturalness*, it consistently receives higher ratings than the baselines across all criteria. A Bonferroni post-hoc test further confirms that our model significantly outperforms all baselines in *Coherence* and *Musicality*. These results align with the objective evaluation and demonstrate that our model captures music style more effectively and produces coherent, high-quality piano cover arrangements.

## 5.5 EVALUATION ON AUDIO-TO-SYMBOLIC ALIGNMENT

While the Q-Former bridges the modality gap for audio-to-symbolic arrangement, it can also operate independently as an audio-to-symbolic retriever. In this setting, the Q-Former measures stylistic coherence between an audio clip and a symbolic segment by comparing their learned representations. Given an audio query and a set of symbolic candidates, the model can retrieve the symbolic piece that best aligns with the music style of the query audio. To the best of our knowledge, CLaMP3 (Wu et al., 2025) is the only existing model with an audio-to-symbolic alignment capability, and we aim to evaluate whether our approach can achieve superior performance.

### 5.5.1 AUDIO-TO-SYMBOLIC RETRIEVAL

To assess the alignment capability of the Q-Former, we evaluate it after Stage-I training on the audio-to-MIDI retrieval task. In each of 10 independent runs, we construct a test set of 128 pairs of 10-second audio and 4-bar MIDI, randomly sampled from PIAST and POP909 (64 pairs each). For each audio query, the model is tasked with retrieving its corresponding MIDI segment from the full pool of 128 candidates. We consider two evaluation settings: one in which MIDI candidates are randomly transposed to all 12 keys, and the other without transposition. This setup helps us examine the model's robustness in capturing stylistic features beyond absolute pitch and key. Performance is measured using three metrics: *Top-1 Accuracy* (*Acc@1*), *Top-5 Accuracy* (*Acc@5*), and *Mean Rank*. We report the mean and standard error across the 10 resampled runs.

As shown in Table 3, we compare our model against CLaMP3 in addition to a random guessing bot for sanity check. In CLaMP3, audio and MIDI are aligned indirectly via text due to the greater availability of music–text pairs in both domains. While this indirect alignment allows CLaMP3 to perform substantially better than random guessing, its *Acc@1* remains low. Interestingly, transposing the MIDI candidates has a noticeable effect, leading to a 4-point drop in *Acc@5* and a 6-rank

Table 3: Objective evaluation on audio-to-MIDI retrieval. Results are compared against baseline models and across two evaluation settings: MIDI with random transposition (left) and without transposition (right), highlighting robustness in capturing stylistic features beyond absolute pitch and key.

| | w/o Transposition | | | w/ Random Transposition | | |
|---|---|---|---|---|---|---|
| | Acc@1 (%) ↑ | Acc@5 (%) ↑ | Rank ↓ | Acc@1 (%) ↑ | Acc@5 (%) ↑ | Rank ↓ |
| **Random** | $1.4 \pm 0.3$ | $4.8 \pm 0.8$ | $64.4 \pm 1.7$ | $0.7 \pm 0.2$ | $4.2 \pm 0.5$ | $65.4 \pm 1.0$ |
| **CLaMP** | $3.4 \pm 0.4$ | $15.0 \pm 0.6$ | $42.5 \pm 0.2$ | $3.4 \pm 0.4$ | $11.0 \pm 0.6$ | $48.5 \pm 0.3$ |
| **Ours** | $\mathbf{71.4 \pm 1.5}$ | $\mathbf{95.1 \pm 0.5}$ | $\mathbf{2.1 \pm 0.1}$ | $\mathbf{70.2 \pm 1.5}$ | $\mathbf{94.8 \pm 0.5}$ | $\mathbf{2.1 \pm 0.1}$ |

Table 4: Ablation study on individual pre-training objectives for cross-modal alignment.

| | PIAST | | POP909 | |
|---|---|---|---|---|
| | Acc@1 ↑ | Rank ↓ | Acc@1 ↑ | Rank ↓ |
| **C** | $96.7 \pm 0.3$ | $1.8 \pm 0.2$ | $36.2 \pm 1.3$ | $5.2 \pm 0.3$ |
| **C+M** | $\mathbf{97.2 \pm 0.4}$ | $\mathbf{1.6 \pm 0.2}$ | $40.8 \pm 0.9$ | $5.0 \pm 0.4$ |
| **C+M+G** | $97.1 \pm 0.3$ | $1.7 \pm 0.2$ | $\mathbf{44.7 \pm 1.4}$ | $\mathbf{4.7 \pm 0.4}$ |

increase in *Mean Rank*. This is probably because key signatures are frequently referenced in text descriptions of music, making CLaMP3 particularly sensitive to pitch-level features while struggling to capture finer stylistic nuances. In comparison, our model consistently outperforms CLaMP3 across all metrics and exhibits negligible performance differences between the transposed and non-transposed settings. This demonstrates that our Q-Former learns more robust audio-to-symbolic alignments, effectively capturing stylistic coherence beyond surface-level attributes.

### 5.5.2 Ablation Study on Pre-Training Objectives

We are also interested in the contribution of each pre-training objective in Section 3.3 to cross-modal alignment. We conduct an ablation study on the Q-Former's audio-to-MIDI retrieval performance based on three different pre-training configurations: contrastive loss only (**C**), contrastive + matching losses (**C+M**), and contrastive + matching + generative losses (**C+M+G**). As in the previous section, we repeat our experiment over 10 independent runs on 128 resampled audio-MIDI pairs.

As shown in Table 4, we conduct evaluation separately on PIAST and POP909. The former involves piano-only music, while the latter includes multi-instrumental accompaniments, requiring the model to extract style from richer audio textures. We observe that the performance difference is relatively small on PIAST, suggesting that contrastive learning alone may suffice for simpler piano alignment. However, on POP909, we see both the matching and generative losses contribute meaningfully to improved retrieval accuracy and lower mean rank. These findings indicate that all three objectives are important for learning robust, generalizable cross-modal alignment.

## 6 Conclusion

In this paper, we introduce a cross-modal framework for audio-to-symbolic arrangement. By repurposing the Q-Former to align audio and symbolic modalities, our model extracts and applies implicit music style using pre-trained music LMs, enabling expressive piano arrangement conditioned on both a lead sheet and an audio reference. Through a two-stage training process—combining representation learning and generative modeling—we extract stylistic features from a frozen, large audio LM and guide a symbolic LM without re-training either backbone. We conduct quantitative experiments on piano cover generation and provide qualitative demos of style transfer. Results demonstrate improved audio-to-symbolic coherence and musicality, highlighting the potential of this framework for controllable, style-aware music generation beyond explicitly labeled content.

## 7 Ethics Statement

We confirm that our work adheres to the ICLR Code of Ethics. Datasets used in this study are open-source with appropriate licenses: the POP909 dataset under MIT license and the PIAST dataset

under CC-BY-NC 4.0 license. Our subject evaluation was conducted through online crowdsourcing, which bears minimal risk. Participants were informed that participation was voluntary and that they could withdraw at any time without negative consequences. No personally identifiable information was collected. Further details of the subjective evaluation setup are described in Appendix D.

## 8 REPRODUCIBILITY STATEMENT

We have made every effort to ensure the reproducibility of our results. Section 4 provides a detailed description of our experimental setup, including the datasets (Section 5.1) and model configurations (referred to Appendix A). Note that, due to copyright restrictions, the audio portion of POP909 is not publicly distributed. We obtained it by reaching out to the original authors. For objective experiments, we repeated each experiment 10 times and report the mean and standard error of the mean. For subjective evaluation, we applied within-subject ANOVA followed by post-hoc paired t-tests. To further facilitate reproducibility, we will release our implementation code upon publication.

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

## A  MODEL CONFIGURATION AND TRAINING DETAILS

Our model comprises three components: an audio LM, a symbolic music LM, and a Q-Former connecting the two. This section provides detailed configurations of each component module.

### A.1  AUDIO LM

We use MusicGen-Large (Copet et al., 2023) as our audio LM. We discard the text encoder and retain only the music decoder, a 48-layer Transformer. Audio codecs are fed to the decoder and we extract the hidden representations from the 25th layer, as prior probing studies (Wei et al., 2024; Ma et al., 2024; Vásquez et al., 2024; Castellon et al., 2021) suggest that middle layers capture more musically meaningful features. This setup retains 1.7B frozen parameters from MusicGen.

### A.2  SYMBOLIC MUSIC LM

For symbolic music arrangement, we adopt MuseCoco-xLarge (Lu et al., 2023), which is a 24-layer Transformer decoder pre-trained on large-scale symbolic music corpora. We remove its text-related components and keep 1.2B frozen parameters from the music decoder.

### A.3  Q-FORMER

The Q-Former comprises 186M learnable parameters, which is significantly smaller than the billion-scale backbone models. In Stage-I, it is pre-trained in FP16 using batch size 128 for 10 epochs (130K iteration). The LoRA adaptor in Stage-II adds 5M parameters and we fine-tune the model for another 5 epochs using batch size 32. Both training stages are conducted on four RTX A40 GPUs (48GB each). We use the AdamW optimizer (Loshchilov & Hutter, 2019) with an initial learning rate of 1e-4, a linear warm-up over the first 1k steps, and a cosine decay schedule to a final rate of 1e-5. At test time, we use top-$k$ sampling with $k = 15$.

## B  OBJECTIVE METRICS

We introduce five statistical metrics to evaluate *content preservation* and *style coherence* for the piano cover generation tasks. This section provides the definitions of each metric.

### B.1  MELODY CHROMA ACCURACY (MCA)

We use the `melody.raw_chroma_accuracy` metric[5] provided by mir_eval (Raffel et al., 2014) to evaluate the similarity between two monophonic melody sequences. For the reference melody, we

---

[5]https://mir-eval.readthedocs.io/latest/api/melody.html#mir_eval.melody.raw_chroma_accuracy

apply Demucs (Rouard et al., 2023) to isolate the vocal stem from the audio and then extract the F0 contour using pYIN (Mauch & Dixon, 2014) provided by librosa (McFee et al., 2015). For the estimated melody, we obtain the melody skyline (Uitdenbogerd & Zobel, 1998) from the generated piano cover MIDI and convert the MIDI pitches to frequencies. The two melodies are compared position-wise under a tolerance of 50 cents. While this implementation follows (Tan et al., 2024a), we additionally ensure that the two melodies are temporally aligned to the same sequence length.

## B.2 CHORD ACCURACY (CA)

We introduce *Chord Accuracy* from (Ren et al., 2020; Zhao et al., 2024) to measure the similarity between two chord sequences. For the reference sequence, when annotated chords are not available, we apply the method of (Jiang et al., 2019) to detect chords from the input audio. For the estimated sequence, we use (Jiang, 2025) to detect chords from the generated piano cover MIDI. Both chord sequences are aligned and compared at 1-beat granularity in terms of root and full quality based on the MIREX TeTrads rule.[6]

## B.3 GROOVE PATTERN COHERENCE (GPC)

*Grooving Pattern Coherence* (*GPC*) evaluates the *grooving pattern* similarity between the generated piano cover and human's arrangement. The *grooving pattern*, which is defined in (Wu & Yang, 2020), represents the positions in a MIDI segment at which there is at least one note onset. We consider 4-bar segments at 1/4-beat granularity, deriving *grooving pattern* $\mathbf{g}$ as a 64-dimensional binary vector. The *GPC* over a test piece is defined as follows:

$$GPC = \frac{1}{N} \sum_{n=1}^{N} \cos(\mathbf{g}_n^{\mathrm{hm}}, \mathbf{g}_n^{\mathrm{cv}}), \tag{1}$$

where $N$ is the number of non-overlapping 4-bar segments in the test piece. $\cos(\cdot, \cdot)$ computes the cosine similarity. $\mathbf{g}^{\mathrm{hm}}$ and $\mathbf{g}^{\mathrm{cv}}$ represent the *grooving pattern* feature from the human arrangement and the generated piano cover, respectively. The *GPC* metric is omitted for evaluation on Ballroom/GTZAN, where human's piano arrangement is not available.

## B.4 VELOCITY CONTOUR COHERENCE (VCC)

*Velocity Contour Coherence* (*VCC*) evaluates the similarity of the *velocity contour* between the generated piano cover and the human arrangement, using the same formulation as *GPC*. We define the *velocity contour* as a time-series feature representing the average note velocity at each timestep. It has the same dimensionality and temporal granularity as the *grooving pattern*, but instead of a binary vector, it consists of real values ranging from 0 to 128. The *VCC* metric is omitted for Ballroom/GTZAN, where human's piano arrangement is not available.

## B.5 TEMPO ACCURACY (TA)

*Tempo Accuracy* (*TA*) evaluates the correctness of the estimated tempo $\mathrm{tp}^{\mathrm{est}}$ relative to the reference (ground-truth) tempo $\mathrm{tp}^{\mathrm{ref}}$. We first define correctness indicator $D(k)$ for one test piece as follows:

$$D(k) = \mathbb{1}\left(\frac{|\mathrm{tp}^{\mathrm{ref}} - k \times \mathrm{tp}^{\mathrm{est}}|}{\mathrm{tp}^{\mathrm{ref}}} < 0.08\right), \tag{2}$$

where $\mathbb{1}(\cdots)$ is the indicator function. $k \in \{1/2, 1, 2\}$ takes into account the octave ambiguity. The tolerance threshold of 0.08 follows the empirical setting in mir_eval.[7]

We then define tempo accuracy *TA* as:

$$TA = \max\left(D(1),\ 0.5 \cdot D(1/2),\ 0.5 \cdot D(2)\right), \tag{3}$$

---

[6]https://mir-eval.readthedocs.io/latest/api/chord.html#mir_eval.chord.tetrads
[7]https://mir-eval.readthedocs.io/latest/api/tempo.html#mir_eval.tempo.detection

Table 5: Objective evaluation on audio-to-symbolic style transfer. Results are reported by mean ± standard error. The star (*) indicates significant differences ($p < 0.05$) based on the Friedman test.

| | MCA | CA | GPC | VCC | TA |
|---|---|---|---|---|---|
| **Ours** | **28.9**±0.5* | **40.0**±1.0* | **64.0**±1.1 | **62.8**±1.1 | **73.8**±2.0* |
| **w/o PT** | 25.1±0.5 | 33.2±1.0 | 62.9±1.3 | 62.0±1.3 | 69.2±2.1 |

where we consider half-tempo and double-tempo matches as partially correct (weight 0.5). This is because tempo perception is known to exhibit octave ambiguity, where perceptions at multiple metrical levels are still considered valid rather than true perceptual errors (Dixon, 2001).

We derive the ground-truth tempo from beat annotations when available. On Ballroom/GTZAN, we use the audio tempo estimation results by madmom (Böck & Davies, 2020), and we omit the *A2M* baseline in this setting because it already relies on madmom's estimation in its pipeline.

## C   EVALUATION ON CROSS-MODAL STYLE TRANSFER

To further evaluate our model's style transfer capability, we note that, to the best of our knowledge, our method is the first to enable cross-modal audio-to-symbolic style transfer, and thus no established baseline exists for direct comparison. Nevertheless, we conduct an objective ablation study comparing our full model (*Ours*) against a variant without Stage-I pretraining (*w/o PT*), to demonstrate the validity of our two-stage training paradigm for cross-modal style transfer. For this experiment, we collect content lead sheets from the POP909 test split. In 10 independent trials, each lead sheet is paired with a style audio reference randomly drawn from the PIAST test split. This yields a total of 450 style-transfer pairs, and both our full model and the ablation variant generate outputs for each pair. We evaluate *content preservation* using *MCA* and *CA*, which measure melody/chord similarity to the input lead sheet, and *style coherence* using *GPC*, *VCC*, and *TA*, which measure groove/velocity/tempo alignment with the style reference audio's transcription.

As shown in Table 5, our full model outperforms the *w/o PT* variant across all metrics with a good margin, indicating stronger style-transfer capability in both *content preservation* and *style coherence*, thereby validating our two-stage training paradigm. This finding also aligns with the subjective evaluation results in Section 5.4, where our full model is consistently preferred in terms of *Audio-to-Symbolic Coherence*, in addition to other musicality criteria.

## D   SUBJECTIVE EVALUATION DETAILS

Our subjective evaluation is conducted through an online crowdsourcing study, where participants complete a survey consisting of listening and rating tasks. This section provides additional details on the survey design and participant profile.

### D.1   GENERAL INSTRUCTIONS

Participants receive the following general instructions, which clarify their rights and the conditions of participation:

- Participation is entirely voluntary, and one may withdraw at any time without any negative consequences.
- No personally identifying information is collected; all responses are anonymous and used solely for research purposes.

### D.2   SURVEY DESIGN

Our survey consists of 6 pages, each presenting 4 versions of piano cover arrangements corresponding to a common test audio piece. The test audio pieces are drawn from the Ballroom and GTZAN datasets and span a variety of genres. The 4 arrangement versions are produced by our model and all

baseline models (*PCG2*, *A2M*, and *w/o PT*), respectively. For each model, we select the best result from 3 independently generated samples to avoid occasional low-probability failures (e.g., incomplete or degenerate outputs) from disproportionately affecting subjective evaluation. All models to be evaluated are anonymized, and their orders on each page are randomized.

We acknowledge that participants may use different internal criteria when evaluating music. To minimize such variability, we conduct within-subject ANOVA (Scheffe, 1999) for data analysis, which ensures that variances in ratings reflect differences among the models rather than participants. To control response quality, we only accept complete evaluation sets; that is, participants have to rate all four models under a common test piece for their responses to be considered valid.

### D.3 COMPLETION TIME

Each participant is randomly assigned 3 out of the 6 pages. On each page, participants first listen to the test audio piece, and then listen to and rate 4 corresponding piano cover samples. All samples in the survey are 16 bars long and rendered to audio using the Cakewalk TTS-1 soundfont, producing approximately 40 seconds of audio per sample. This design targets a total completion time of 10–15 minutes, ensuring that participants have sufficient time to listen carefully without excessive fatigue. The actual completion time observed on average is 12 minutes.

### D.4 PARTICIPANT PROFILES

Participants are asked to self-identify their musical background as *amateur*, *intermediate*, or *professional*, following the guidelines below:

- **Amateur**: I enjoy listening to music. I can play/sing/compose short music pieces. I know a little music theory. I can evaluate a composition based on my feelings.
- **Intermediate**: I have some experience in performing, composing, or other music activities. I know a certain amount of music theories that can help me evaluate a composition.
- **Professional**: I am now pursuing/have completed a music degree, or having equivalent background. I am proficient in using music theory to evaluate a composition.

Among the 21 valid responses collected, 6 participants identified as *amateur* (28.6%), 11 as *intermediate* (52.4%), and 4 as *professional* (19.0%). All authors are excluded from the survey.

## E LIMITATION

Our proposed method demonstrates the ability to learn implicit music style from audio. At the current stage of this work, we acknowledge that the extracted style primarily represents *segment-level global* characteristics. Here, "global" refers to a latent style profile at the level of short 4-bar segments, which is still considerably more fine-grained (and more local) than typical global attributes such as genre labels and textual style tags. Our underlying assumption is that music style remains locally consistent at the segment or bar level. We empirically found that this assumption holds across most music traditions, which guarantees our model to perform reliably under this design. While longer generation can be achieved using windowed sampling, we acknowledge that this approach may smooth over intended stylistic transitions at phrase boundaries, thus leading to diminished expressivity at longer timescales. When considering inter-phrase and longer-term music development, we recognize that a single segment-level style representation is insufficient to capture evolving dynamics. Also, subject to the availability of audio-symbolic data, this work is dedicated to piano arrangement. The cross-modal arrangement of long-term, multi-track music may require hierarchical or temporally adaptive style modelling, which is an important direction for our future work.

## F ACKNOWLEDGMENT OF LLM USAGE

We acknowledge the use of OpenAI's ChatGPT to refine the writing in this paper. The tool was employed to improve coherence, grammar, and readability. All research ideas, experimental designs, analysis, and conclusions are the authors' own.

