# OpenReview forum: "Learning Music Style For Piano Arrangement Through Cross-Modal Bootstrapping"
_ICLR.cc/2026/Conference — Submitted to ICLR 2026_

### Official Review · Reviewer_um9i · 2025-10-24

**Soundness:** 3
**Presentation:** 3
**Contribution:** 2
**Rating:** 6
**Confidence:** 4

**Summary:**

This paper proposes a cross-modal framework to learn and apply implicit music styles from raw audio to symbolic music generation. The model uses a Querying Transformer (Q-Former) to extract style representations from a pre-trained audio language model and conditions a symbolic language model to generate piano arrangements. A two-stage training strategy is adopted: (1) contrastive learning aligns auditory style with symbolic expression; (2) generative modeling performs arrangement. The system generates piano performances conditioned on both a lead sheet (content) and a reference audio example (style), enabling controllable, stylistically faithful results. Experiments show significant gains in piano cover generation, style transfer, and audio-to-MIDI retrieval.

**Strengths:**

The main strengths of the paper are:
- The proposed system obtains good results in style transfer, and this with audio input of diverse styles and instrumentations.
- The experimental plan is solid and well highlights the merits of the proposed system.
- The paper is clearly and nicely written
- The (online) demo is very convincing

**Weaknesses:**

The main weaknesses of the paper are:
- The novelty of the proposed system mainly relies on existing building blocks ;
- The overall proposed system is quite complex in the overall and it is not clear that it is better or more controllable than a much simpler system based on a style recognition from audio (or style retrieval/classification from audio) and then audio generation from lead-sheets (e.g. current music generation systems using chords sequence and a style label as inputs can provide excellent rendering in hundreds of styles).
- The design of the perceptual test is lacking some details.

**Questions:**

-	Perceptual tests: the participants participated online and had diverse musical backgrounds. More details could be given: Were they all musicians ? capable or reading lead sheets ? were the authors excluded from the survey ? how the quality of participant’s answers was controlled/monitored ?
-	Objective evaluation: in section 5.3, it is mentioned that “..Clamp3 compuutes similarity […] and all generated symbolic piano covers”.  Do you mean here all covers from all audio of the database with 1 cover per audio ? How many covers then (to have an idea of how good/bad is a rank of 16 or so on POP909) ?
-	The generative objective for the Q-former is not clear. It is said (p4) that the Q-former is trained on an input audio and lead sheet (but the lead lead is not input to the Q-former in stage 1 in figure 1 and figure 2… this is confusing and it is not really clear what is finally used as input information for stage 1.

---

> ### Author Response · Authors · 2025-11-20
>
> Thank you for your review and feedback! Please allow us to first respond to the points raised in Weakness and Questions. We hope this could address your concerns. We will then comprehensively incorporate these clarified ideas into our manuscript to provide sufficiently clear methods and evaluation settings.
>
> **W1. The novelty of the proposed system**
>
> We would like to clarify that our main novelty lies in addressing the audio-to-symbolic arrangement problem through a new and creative lens of cross-modal style transfer. Although our system leverages existing building blocks, the way these components are adapted is methodologically original. Our approach effectively bridges a large, frozen audio LM to a symbolic LM, enabling cross-modal, style-aware arrangement without the substantial computational cost required to retrain large foundation models. The choice and combination of these modules is therefore not a simple aggregation of existing parts, but a purposeful design tailored to solving a challenging cross-modal style-transfer task. With the growing availability of large-scale LMs in both audio and symbolic music modalities, we believe this approach offers practical insights for multi-modal music generation.
>
> **W2. Comparing to a simpler system based on style recognition**
>
> We appreciate the reviewer’s concern and would like to clarify why our approach goes beyond what a simpler “style-recognition + conditional generation” pipeline can offer. Conventional style-recognition models typically output a categorical label (e.g., genre, mood) or a set of coarse textual attributes. Such representations inevitably abstract away the rich implicit style cues contained in the audio, such as groove patterns and performance dynamics. When these coarse labels are used to condition a symbolic or audio generator, the result cannot faithfully reproduce the detailed stylistic nuances from the reference audio. In contrast, our method performs cross-modal representation learning to align the latent audio and symbolic space directly. This allows the model to "hear" the finer-grained, segment-level style features and transfer them without the information bottleneck that simpler pipelines suffer from.
>
> **W3. & Q1. Details on the perceptual tests**
>
> Thank you for your question regarding the perceptual test. We will include additional details in the revised manuscript. Here, we would like to briefly illustrate the music background of participants and how we control the quality and fairness of subjective comparison.
>
> Participants were asked to self-identify their musical background as *amateur*, *intermediate*, or *professional*, following the guidelines below:
> * ***Amateur***: I enjoy listening to music. I can play/sing/compose short music pieces. I know a little music theory. I can evaluate a composition based on my feelings.
> * ***Intermediate***: I have some experience in performing, composing, or other music activities. I know a certain amount of music theories that can help me evaluate a composition.
> * ***Professional***: I am now pursuing/have completed a music degree,  or having equivalent background. I am proficient in using music theory to evaluate a composition.
>
> Among the 21 participants, 6 identified as *amateur* (28.6%), 11 as *intermediate* (52.4%), and 4 as *professional* (19.0%). All authors were excluded from the survey.
>
> We acknowledge that participants may use different internal criteria when evaluating music. To minimize such variability, we employed a within-subject design, where each participant evaluated all four methods for every test piece. For the same reason, we conducted within-subject ANOVA for data analysis, which ensures that differences in ratings reflect the methods rather than variations among participants. To control response quality, we only accepted complete evaluation sets; that is, participants had to rate all four methods for a piece for their responses to be considered valid.

---

> ### Author Response · Authors · 2025-11-20
>
> **Q2. Details on objective evaluation using CLaMP3 in Section 5.3**
>
> The objective evaluation in Section 5.3 was conducted on two test sets: the POP909 test split (45 pieces) and Ballroom/GTZAN out-of-distribution set (100 pieces). In each of 10 independent runs, our model and each baseline generated one (symbolic) piano cover per (audio) piece. Under this setting, CLaMP3 computes *Retrieval Accuracy* and *Mean Rank* to indicate audio-to-symbolic coherence. We report both the mean and the standard error over the 10 runs to ensure reliability and repeatability of the results. A random guessing bot would produce a mean rank of 22.5 on POP909, and 50 on Ballroom/GTZAN.
>
> We also refer the reviewer to our responses to reviewers PuRH and sCao, where we have introduced additional objective metrics and supplemented evaluations. These evaluations more rigorously validate our approach in piano cover generation and audio-to-symbolic style transfer, thus further strengthening the soundness of our method.
>
> **Q3. Q-Former input during Stage-1 training**
>
> We appreciate the reviewer’s comment and will clarify the Q-Former inputs used during Stage-1 training. The Q-Former indeed operates under two different input configurations, corresponding to the different objectives in Stage 1:
>
> 1. **Contrastive loss and matching loss**
>
>       For these two objectives, the Q-Former input is:
>
>       * **\<query embeddings\>** \<s\> **\<piano arrangement tokens\>** \</s\>
>
>      In this setting, the query embeddings are trained to capture general information shared between the audio and symbolic modalities, since the contrastive and matching objectives explicitly align audio and symbolic latent spaces.
>
> 2. **Audio-grounded generation loss**
>
>       For this objective, the Q-Former input is:
>
>       * **\<query embeddings\>** **\<lead sheet tokens\>** \<DEC\> **\<piano arrangement tokens\>** \</s\>
>
>      Here, the \<DEC\> token signifies a decoding task. The lead sheet tokens serve as the content condition. This design intentionally constrains the query embeddings to focus on style. Since the lead sheet already provides the content (melody + chord), the Q-Former must extract style-related representations from the audio latent to reconstruct the piano arrangement.
>
> **Manuscript Revision**
>
> Thank you again for your review and feedback! We are currently working to incorporate the supplemented experimental results and clarified explanations into the revised version of our manuscript. If you have any further questions or concerns, please do not hesitate to let us know!

---

> > ### Comment · Reviewer_um9i · 2025-11-25
> >
> > Thank you authors for the detailed rebuttal and answers to the points raised in the review. I would like to confirm my initial positive evaluation score.

---

### Official Review · Reviewer_sCao · 2025-10-31

**Soundness:** 2
**Presentation:** 3
**Contribution:** 3
**Rating:** 6
**Confidence:** 3

**Summary:**

The paper proposes a cross-modal framework that learns implicit musical style from raw audio and applies it to symbolic music generation. A two-stage model is introduced: Stage 1 performs audio-symbolic representation learning via three complementary losses (contrastive, matching, and audio-grounded generative) to distill style information with a Q-Former.
Stage 2 conditions a symbolic LM (MuseCoco) on both the learned style embedding and a lead sheet for controllable piano arrangement. The system supports multiple tasks including piano-cover generation, style transfer, and audio-to-MIDI retrieval.

**Strengths:**

1. The paper defines a new setting — learning implicit style from audio for symbolic arrangement — that meets the need for fine-grained style control, distinguishing it from transcription or text-conditioned generation.
2. The two-stage design enables flexible applications (piano cover generation, style transfer, and audio-to-MIDI retrieval).
3. The three complementary training losses effectively achieve audio-symbolic representation learning; retrieval results confirm that the learned style representation is robust and generalizable.
4. The presentation is clear and visually well-organized.

**Weaknesses:**

1. While audio input allows more detailed style control, the current framework reduces usability — users must provide a reference audio rather than a simple text prompt. It would be more practical to retain MuseCoco’s original text-conditioning ability for flexible use.
2. The learned style representation mainly captures global characteristics and fails to model segment-level or bar-level dynamic style variations.
3. Objective evaluation is too limited. Using only CLaMP3 as a metric is insufficient to justify the quality and style consistency of generated outputs. Piano cover generation should be evaluated with richer metrics. In addition, the style transfer capability should be further validated through subjective listening evaluations, demonstrating perceptual differences and user preference across styles.
4. Minor: In Table 2, the Rank column should have a downward arrow (↓) instead of upward.

**Questions:**

See Weakness section above.

---

> ### Author Response · Authors · 2025-11-20
>
> Thank you for your review and insight! We appreciate your generous compliment about the innovation of this work. In the meantime, we recognize the need for a more thorough evaluation of piano cover generation and style transfer. We have designed objective metrics and conducted new experiments to complement our existing results. We provide details as follows and hope these address your concern.
>
> **W1. Integrating audio and text control**
>
> We fully recognize the practicality and flexibility of text-based conditioning, and we agree that integrating both audio and text controls could further broaden usability. In this paper, we focus on a more specific audio-guided arrangement setting. Apart from piano cover arrangement, our model enables rapid mock-ups when the user imagines how an auditory style A would apply to a composition B. Such cross-modal style-transfer capability can further foster creative exploration and artistic expression. We will explore the integration of audio and text instructions for music arrangement in our future work.
>
> **W2. Regarding “global” characteristics of style representation**
>
> This weakness aligns with our discussion in the Limitation section (Appendix B, Line 696). We would like to clarify that the “global” here refers to “segment-level global”, which is still considerably more fine-grained (and more local) than typical global attributes such as genre and textual style tags. Our underlying assumption is that musical style remains locally consistent at the segment or bar level. We empirically found that this assumption holds across most music traditions, which guarantees our model to perform reliably under this design.
>
> At the same time, we acknowledge that this design may smooth over intended stylistic transitions at phrase boundaries, leading to diminished expressivity at longer timescales. When considering inter-phrase and longer-term music development, we recognize that a single segment-level style representation is insufficient to capture evolving dynamics. Addressing this limitation may require hierarchical or temporally adaptive style modelling, which is an important direction for our future work.
>
> **W3. Supplemented objective evaluation for piano cover generation**
>
> We recognize that relying solely on CLaMP3 would be insufficient for a rigorous evaluation. To assess the piano cover generation task more comprehensively, we introduce a suite of additional objective metrics. These metrics capture both content preservation and style coherence, and are summarized below:
> * **Content preservation metrics**
>   * Melody Chroma Accuracy ***(MCA)***
>   * Chord Accuracy ***(CA)***
> * **Style coherence metrics**
>   * Grooving Pattern Coherence ***(GPC)***
>   * Velocity Contour Coherence ***(VCC)***
>   * Tempo Accuracy ***(TA)***
>
> We refer the reviewer to our response to reviewer PuRH for detailed definitions of these metrics and the corresponding evaluation for the piano cover generation task.

---

> ### Author Response · Authors · 2025-11-20
>
> **(continued) W3. Supplemented objective evaluation for style transfer**
>
> To further evaluate the model’s style transfer capability, we note that, to the best of our knowledge, our method is the first to enable cross-modal audio-to-symbolic style transfer, and thus no established baseline exists for direct comparison. Nevertheless, we conduct an objective ablation study comparing our full model (*Ours*) against a variant without Stage-1 pretraining (*w/o PT*), to demonstrate the validity of our two-stage training paradigm for cross-modal style transfer.
>
> For this experiment, we collect content lead sheets from the POP909 test split. In 10 independent trials, each lead sheet is paired with a style audio reference randomly drawn from the PIAST test split. This yields a total of 450 style-transfer pairs, and both our full model and the ablation variant generate outputs for each pair. We evaluate content preservation using *MCA* and *CA* (measuring melody/chord similarity to the *input lead sheet*), and style coherence using *GPC*, *VCC*, and *TA* (measuring groove/velocity/tempo alignment with the *style reference audio’s transcription*). The results are reported in the form of mean ± standard error. The star (*) indicates significant differences (p < 0.05) based on the Friedman test.
>
>   |        | MCA        | CA         | GPC      | VCC      | TA         |
>   | ------ | ---------- | ---------- | -------- | -------- | ---------- |
>   | **Ours**   | **28.9**±0.5* | **40.0**±1.0* | **64.0**±1.1 | **62.8**±1.1 | **73.8**±2.0* |
>   | **w/o PT** | 25.1±0.5   | 33.2±1.0   | 62.9±1.3 | 62.0±1.3 | 69.2±2.1   |
>   | | | | | | |
>
> The evaluation shows that our full model outperforms the *w/o PT* variant across all metrics, indicating stronger style-transfer capability in both content preservation and style coherence, thereby validating our two-stage training paradigm. This finding also aligns with the subjective evaluation results in our paper manuscript, in which our full model is consistently preferred in terms of *Audio-to-Symbolic Coherence*, in addition to *Naturalness*, *Creativity*, and *Musicality*.
>
> **Manuscript Revision**
>
> Thank you again for your review and feedback! We are currently working to incorporate the supplemented experimental results and clarified explanations into the revised version of our manuscript. If you have any further questions or concerns, please do not hesitate to let us know!

---

### Official Review · Reviewer_PuRH · 2025-11-02

**Soundness:** 2
**Presentation:** 2
**Contribution:** 3
**Rating:** 6
**Confidence:** 3

**Summary:**

The article introduces a novel cross-modal framework for generating expressive symbolic piano arrangements by learning implicit music style from raw audio. The primary objective is to generate a symbolic piano arrangement that is conditioned on two inputs: an audio example (for the style) and a lead sheet (for the melody and harmony content). This enables controllable and stylistically faithful arrangements that go beyond explicit text labels. The core of the framework is a Q-Former, inspired by BLIP-2, which acts as a lightweight bottleneck to bridge the gap between a frozen, pre-trained audio Language Model (LM) and a symbolic music LM. The Q-Former is designed to extract a representation of the implicit music style from the audio LM's hidden states. Similar to BLIP-2, a two-step sequential training strategy is used to "bootstrap" the system without retraining the large LM backbones. First, in a cross-modal representation learning, the Q-Former is trained using three complementary objectives (Contrastive Learning, Audio-Symbolic Matching, and Audio-Grounded Symbolic Generation) to align the auditory style representation with symbolic expression, while disentangling it from explicit music content. Then, in the second step, the symbolic LM (MuseCoco with a LORA adapter) is conditioned on the extracted style embedding from the Q-Former and the lead sheet content to generate the final piano arrangement.

**Strengths:**

- The paper tackles an ambitious and important problem in symbolic music generation: disentangling and transferring music style from a complex audio signal to a symbolic arrangement. By defining style as "hidden in concrete music examples" rather than relying on abstract text labels, the authors set a high bar, aiming for expressive performance nuances rather than just generic genre characteristics.

- The central technical contribution—using a Querying Transformer (Q-Former) to bridge a frozen Audio Language Model (LM) and a frozen Symbolic LM (MuseCoco)—is highly innovative and resource-efficient.

- The framework allows for an interesting form of controllable music generation. The final arrangement is jointly conditioned on two independent inputs: 1. The explicit lead sheet (melody and chords), and 2. The implicit style from a raw audio reference. This dual control is superior to systems that rely on simple text tags or require the arrangement to be fully contained within the style reference, making the model flexible for diverse creative applications.

- Audio examples reveal partially successful transfer of musical style (notably tempo, intensity contours for the main melody), which would probably partly qualify as implicit style and partly as arrangement.

- The framework's intermediary component, the Q-Former, is successfully evaluated in an experimental setup requiring the retrieval of the correct MIDI excerpt given audio. While the number of candidates remains small, it still indicates a successful extraction of the properties of symbolic music from audio examples.

**Weaknesses:**

- The paper does not define the term "musical style",  the introduction introduces the concept of implicit style. A clear definition of this term remains missing. Authors explain: *When musicians learn a style, instead of relying on abstract definitions like “romantic” or “jazz” alone, they absorb patterns from music examples that share common stylistic traits.* But jazz is not a *definition*, it is a *label* for a collection of examples (all music carrying the label given by humans) anyway. Then there is the interpretation style within the larger genre style. Each artist has their own style that they use to express their personal identity. Here, for piano performance, precise timing, arpeggios, velocity nuances, evolution (e.g., accentuation), and duration variations are certainly important. All this can be described explicitly without too much effort. All this should have been described more thoroughly.

- The evaluation appears weak. I don't quite understand what was measured in the objective evaluation. Authors write: *For each test audio, CLaMP3 computes the similarity between the audio and all generated symbolic piano covers. If the most similar symbolic candidate corresponds to the one generated from the audio input, we count it as a correct match.* If I see correctly, you have 100 audio examples, generate a cover for each of them, then you compare each generated cover with all audio, and you count as success if the original audio turns out to be the most similar one? If I see correctly, this does not allow comparison of the overall style quality between the various algorithms; it could be that one algorithm has very poor overall similarity but has one feature that it successfully copies from the lead sheet. Then this algorithm, even if overall producing bad results, would turn out best compared to the others, which are less coherent but overall more similar. Then, the subjective test does not evaluate style similarity but only musicality.

**Questions:**

Given the fact that the implicit musical style can be described at least partly with explicit parameters, I wonder whether it would not be possible to create some more objective metrics, as for example tempo, velocity contours over the notes, or similar simple metrics. I noted, for example, that the duration of the piano covers is quite different. A successful cover that copies the style should have keep the  duration, no? Or do you include the tempo from the audio into the lead sheet? Another possible metric could be the presence of all notes from the lead sheet.

---

> ### Author Response · Authors · 2025-11-20
>
> Thank you for your thorough review and constructive feedback! We appreciate your generous compliment about the innovation of this work. In the meantime, we acknowledge that some terms in our paper may not be sufficiently clear (**W1**). We also recognize our shortcomings in providing adequate evaluation for the piano cover generation task (**W2** & **Q1**). We have designed objective metrics and conducted new experiments to complement our existing results. We provide details as follows and hope these address your concerns.
>
> **W1. A concrete interpretation of “implicit style”**
>
> In this paper, implicit style primarily regards the *grooving pattern* and *expressive dynamics (velocity and tempo)* that shape a piano arrangement and performance. These qualities can be nuanced and thus hard to precisely convey through explicit labels, but are readily perceivable from concrete audio examples. While such implicit style may span broader compositional and expressive dimensions, we focus on three aspects: *grooving pattern*, *velocity contour*, and *tempo*, and our supplemented experiments (see below) will focus on evaluating style coherence regarding these aspects.
>
> **W2. & Q1. Supplemented objective metrics**
>
> We recognize that CLaMP3 computes a general similarity score between audio input and symbolic output. Therefore, as the reviewer points out, this result alone may not be entirely sufficient to interpret the style quality. To evaluate the piano cover generation task more rigorously, we conduct an additional experiment with newly introduced metrics. These metrics are divided into two categories: 1) evaluating *content preservation* (melody and chord), and 2) evaluating *style coherence*. These metrics are defined as follows:
>
> * **Content preservation metrics**
>
>   * ***Melody Chroma Accuracy (MCA)*** evaluates the similarity between the vocals F0 of input audio and the melody skyline of the piano cover. Our implementation of *MCA* follows [1].
>
>   * ***Chord Accuracy (CA)*** evaluates the harmonic similarity between the generated piano cover and the annotated chord. On Ballroom/GTZAN test set, where chord annotation is not available, we use the audio chord estimation results by [2].
> * **Style coherence metrics**
>   * ***Grooving Pattern Coherence (GPC)*** evaluates the grooving pattern similarity between the generated piano cover and human's arrangement. The grooving pattern, which is defined in [3], is a 1-D time sequence indicating note onset density. This metric is applied to POP909 test set only, where human's piano arrangement is available.
>   * ***Velocity Contour Coherence (VCC)*** evaluates the velocity contour feature between the generated piano cover and human's arrangement. The velocity contour is defined as a 1-D time sequence with average note velocity at each timestep.  This metric is applied to POP909 test set only, where human's piano arrangement is available.
>   * ***Tempo Accuracy (TA)*** evaluates the tempo correctness between the generated piano cover and the ground-truth. On POP909, ground-truth tempo is calculated from the beat annotation. On Ballroom/GTZAN test set, we use the audio tempo estimation results by madmom [4], and we omit the A2M baseline here because it uses madmom's estimation in its pipeline.
>
> [1] C.-P. Tan, et al "Picogen2: Piano cover generation with transfer learning approach and weakly aligned data," in ISMIR 2024.
>
> [2] J. Jiang, et al. "Large-vocabulary chord transcription via chord structure decomposition," in ISMIR 2019.
>
> [3] S.-L. Wu, et al. "The jazz transformer on the front line: Exploring the shortcomings of ai-composed music through quantitative measures," in ISMIR 2020.
>
> [4] S. Böck, et al. "Deconstruct, analyse, reconstruct: How to improve tempo, beat, and downbeat estimation," in ISMIR 2020.

---

> ### Author Response · Authors · 2025-11-20
>
> **(continued) W2. & Q1. Supplemented objective evaluation results**
>
> Across these metrics, a higher value indicates better quality in terms of either content or style. We conduct experiments using the same setup as described in our paper manuscript. Specifically, we run our method and baseline models (*PCG2*, *A2M*, and *w/o PT*) at each test piece in 10 independent rounds, deriving 450 sets of piano cover generation samples for the POP909 (in-distribution) test split, and 1000 sets for Ballroom/GTZAN (out-of-distribution).  The evaluation results are reported in the form of mean ± sem$^s$ (in percentage), where sem is the standard error of mean. Different superscript letters $s$ within a column indicate significant differences (p < 0.05/6) based on Wilcoxon signed rank test with Bonferroni correction.
>
> * **POP909 (In-Distribution)**
>
>   |        | MCA      | CA       | GPC      | VCC      | TA       |
>   | ------ | -------- | -------- | -------- | -------- | -------- |
>   | **Ours**   | 32.0±0.5$^b$ | 33.3±1.1$^b$ | **79.2**±0.5$^a$ | **76.6**±0.5$^a$ | **83.6**±1.5$^a$ |
>   | **PCG2**   | **39.3**±0.6$^a$ | **40.0**±0.9$^a$ | 78.4±0.6$^a$ | 73.2±0.5$^b$ | 74.4±2.0$^c$ |
>   | **A2M**    | 15.0±0.3$^c$ | 22.4±0.9$^d$ | 69.0±0.7$^b$ | 68.6±0.8$^c$ | 77.8±1.4$^b$ |
>   | **w/o PT** | 30.0±0.6$^b$ | 28.8±1.0$^c$ | 78.6±0.5$^a$ | 76.3±0.5$^a$ | 68.6±1.9$^d$ |
>   |   | |   |   |  |   |
>
> * **Ballroom/GTZAN (Out-Of-Distribution)**
>
>   |        | MCA      | CA       | TA       |
>   | ------ | -------- | -------- | -------- |
>   | **Ours**   | **17.8**±0.3$^a$ | **16.3**±0.4$^a$ | **79.4**±1.1$^a$ |
>   | **PCG2**   | 17.2±0.3$^a$ | 15.5±0.5$^b$ | 57.6±1.5$^c$ |
>   | **A2M**    | 10.9±0.2$^b$ | 14.8±0.5$^b$ | -    |
>   | **w/o PT** | 17.2±0.3$^a$ | 15.0±0.4$^b$ | 74.5±1.3$^b$ |
>   |   | |   |   |
>
> While our method leads in most metrics, we find that on POP909, it achieves a lower content score (*MCA* and *CA*) than the *PCG2* baseline. This is probably because our arrangement process relies on Sheetsage-extracted lead sheets, where wrongly estimated chords or melody notes at this stage can propagate to our final results. However, on the out-of-distribution Ballroom/GTZAN dataset, our model surpasses *PCG2* in *MCA* and *CA*, suggesting that *PCG2* is more tailored to pop music (where chords and melodies are relatively simple), whereas our approach generalizes better across diverse music genres. In terms of style coherence, our model outperforms all baselines in *GC* and *VC* on POP909, and in *TA* on both test sets, indicating that it can better preserve the stylistic characteristics from the audio. This finding is consistent with our subjective evaluation, where our model is preferred for its *Audio-to-Symbolic Coherence*, in addition to the other metrics on musicality.
>
> **Manuscript Revision**
>
> Thank you again for your review and feedback! We are currently working to incorporate the supplemented experimental results and clarified explanations into the revised version of our manuscript. If you have any further questions or concerns, please do not hesitate to let us know!

---

### Author Response · Authors · 2025-12-01
**Paper revision uploaded**

Dear reviewers,

Thank you for your comments to improve this paper. We have made a revision, and here is the summary of changes:
* We clarified the idea of "implicit style" in Section 1.
* We added several sentences in Sections 3.2 & 3.3 to better explain our method.
* We supplemented the objective evaluation with new metrics for piano cover generation in Section 5.3.
* We added an ablation study using the new metrics for cross-modal style transfer in Appendix C.
* We provided definitions of the new metrics in Appendix B.
* We provided more details of the subjective evaluation in Appendix D.

**All the changes are highlighted in blue**.

---

### Meta-Review · Area_Chair_YXA1 · 2026-01-07

**Summary:**

The reviewers consider this paper technically sound and creative, but raised concerns about the novelty and evaluation of the proposed cross-modal framework. While the revision addressed several concrete concerns about evaluation, clarity, and ablations, questions remain about whether the proposed cross-modal framework offers a compelling advantage over simpler style-recognition and conditional generation pipelines.

**Reviewer Concerns:**

The rebuttal clarifies the definition of implicit style, expands evals with additional content and style-aware metrics, provides additional ablations, and adds missing methodological details. Some higher-level concerns remain outstanding. It is unclear whether the proposed architecture offers clear advantages over simpler style-recognition–plus–generation pipelines: there is insufficient evidence to support the claims that the learned style representation captures evolving or fine-grained stylistic variation and whether the overall system complexity is warranted given the empirical evidence that has been presented.

**Reviewer Scores:**

PuRH - likely unchanged; conceptual reservations remain.

sCao – likely unchanged; eval concerns mitigated but there are still questions of scope and modeling limitations

um9i – unchanged (explicitly confirmed by um9i).

---

### Decision · Program_Chairs · 2026-01-26

Reject